# Anti-Cancer and Anti-Inflammatory Activities of Three New Chromone Derivatives from the Marine-Derived *Penicillium citrinum*

**DOI:** 10.3390/md19080408

**Published:** 2021-07-23

**Authors:** Yi-Cheng Chu, Chun-Hao Chang, Hsiang-Ruei Liao, Shu-Ling Fu, Jih-Jung Chen

**Affiliations:** 1Institute of Traditional Medicine, School of Medicine, National Yang Ming Chiao Tung University, Taipei 112, Taiwan; chuyc.md07@nycu.edu.tw (Y.-C.C.); slfu@nycu.edu.tw (S.-L.F.); 2Institute of Biopharmaceutical Sciences, Pharmaceutical Sciences, National Yang Ming Chiao Tung University, Taipei 112, Taiwan; changch.ps08@nycu.edu.tw; 3Graduate Institute of Natural Products, College of Medicine, Chang Gung University, Taoyuan 333, Taiwan; liaoch@mail.cgu.edu.tw; 4Department of Pharmacy, School of Pharmaceutical Sciences, National Yang Ming Chiao Tung University, Taipei 112, Taiwan; 5Department of Medical Research, China Medical University Hospital, China Medical University, Taichung 404, Taiwan

**Keywords:** *Penicillium citrinum*, chromone derivatives, anti-inflammatory activity, anti-cancer activity

## Abstract

Three new and uncommon chromone analogs, epiremisporine F (**1**), epiremisporine G (**2**), and epiremisporine H (**3**), were isolated from marine-origin *Penicillium citrinum*. Among the isolated compounds, compounds **2**–**3** remarkably suppressed fMLP-induced superoxide anion generation by human neutrophils, with IC_50_ values of 31.68 ± 2.53, and 33.52 ± 0.42 μM, respectively. Compound **3** exhibited cytotoxic activities against human colon carcinoma (HT-29) and non-small lung cancer cell (A549) with IC_50_ values of 21.17 ± 4.89 and 31.43 ± 3.01 μM, respectively, and Western blot assay confirmed that compound **3** obviously induced apoptosis of HT-29 cells, via Bcl-2, Bax, and caspase 3 signaling cascades.

## 1. Introduction

Marine fungi have been a major source of special structures and bioactive secondary metabolites for lead compounds. In particular, a large number of natural products with biological activities are found in the genus *Penicillium* [1,2,3,4,5,6,7]. For example, the marine fungus *Penicillium citrinum* was found to produce many new bioactive compounds, such as antibacterial dihydroisocoumarins [2], benzopyrans [6], benzophenones [7], antifungal citrinin [4], anticancer benzophenones [7,8], and anti-inflammatory chromone derivatives [8].

Human neutrophils play a significant role in host defenses against pathogen invasion and are the main acute inflammatory mediators [9,10]. After different stimuli, activated neutrophils produce a series of cytotoxins, such as superoxide anion (O_2_^•–^), granule proteases, and bioactive lipids [9,11,12]. Neutrophilic superoxide generation has been linked to many types of inflammation. An inadequately triggered oxidative burst may cause lipid peroxidation, tissue injury, and inflammatory diseases [13].

The main treatment strategies for cancer patients include chemotherapy, operations, and radiotherapy. However, in patients with metastatic cancer, many anti-cancer drugs show limited effects; thus, the development of more effective therapeutic drugs is urgently needed [14].

Undoubtedly, natural products are favorable drug candidates because they are easy to obtain and comparatively safe. Furthermore, natural compounds have been found to be useful to ameliorate the adverse effects of chemotherapeutic agents. Recently, the notion that natural products are an ideal resource for identifying anti-cancer therapeutics has grown globally [15,16,17].

Previously, we reported three new rare chromone analogues, epiremisporine C, epiremisporine D, and epiremisporine E [8]. In this study, we carried out the isolation and structure elucidation of three new compounds, epiremisporine F (**1**), epiremisporine G (**2**), and epiremisporine H (**3**), from the ethanol extract of *Penicillium citrinum*. In addition, we investigated the inhibitory effects of these compounds on superoxide anion generation by fMLP-activated human neutrophils. Moreover, the cytotoxicities of the isolated compounds against two cancer cell lines, colon cancer HT-29 and lung cancer A549, were also examined.

## 2. Results and Discussion

### 2.1. Fermentation, Extraction, and Isolation

In this study, the marine-derived fungal strain *Penicillium citrinum* (BCRC 09F0458) was cultured in solid-state culturing conditions, so as to abound the variability of the fungal secondary metabolites. Chromatographic isolation and purification of the *n*-BuOH-soluble fraction of an EtOH extract of *Penicillium citrinum* on a silica gel column and preparative thin-layer chromatography (TLC) obtained three new compounds (**1**–**3**) (Figure 1).

### 2.2. Structural Elucidation

Compound **1** was isolated as a yellowish amorphous powder. Its molecular formula, C_31_H_26_O_12_, was determined on the basis of the positive HR–ESI–MS ion at *m/z* 613.13294 [M + Na]^+^ (calcd. 613.13219) and supported by the ^1^H and ^13^C NMR data. The IR spectrum showed the presence of hydroxyl (3410 cm^−1^), ester carbonyl (1741 cm^−1^), and conjugated carbonyl (1657 cm^−1^) groups. The ^1^H and ^13^C NMR data of **1** showed the presence of two hydroxy groups, two methyl groups, three methoxy groups, two pairs of meta-coupling aromatic protons, two methylene protons, and three methine protons. The signals at δ 12.19 and 12.42 exhibited two chelated hydroxyl groups with the carbonyl group. Comparison of the ^1^H and ^13^C NMR data of **1** with those of epiremisporine C [8] suggested that their structures were closely related, except that the 2′β-methoxy group of **1** replaced the 2′α-methoxy group of epiremisporine C. This was supported by both HMBC correlations between OMe-2′ (δ_H_ 3.11) and C-2′ (δ_C_ 107.9), and the ROESY correlations between OMe-2′ (δ_H_ 3.11) and H_β_-4′ (δ_H_ 2.87). The relative configuration of **1** was confirmed by the basis of ROESY experiments. The ROESY cross-peaks between H-3/H-4, H-3/H-3′, H-3/H_α_-4′, OMe-2′/H_β_-4′, and H-3/COOMe-2 suggested that H-3, H-4, H-3′, and COOMe-2 are α-oriented, and OMe-2′ is β-oriented. To further confirm the relative configuration of **1**, a computer-assisted 3D structure was obtained by using the molecular-modeling program CS CHEM 3D Ultra 16.0, with MM2 force-field calculations for energy minimization. The calculated distances between H-3/H-4 (2.185 Å), H-3/H-3′ (2.482 Å), OMe-2′/H_β_-4′ (3.412 Å), and H-3/H-16 (2.323 Å) were all less than 4 Å (Figure 2). This was consistent with the well-defined ROESY observed for each of these H-atom pairs (Figure 2). The absolute configuration of **1** was evidenced by the CD Cotton effects at 208.0 (Δ*ε* +13.40), 230.0 (Δ*ε* –5.94), 258.5 (Δ*ε* +19.29), 288.5 (Δ*ε* –7.49), and 330.0 (Δ*ε* +5.40), in analogy with those of epiremisporine E [8]. The ^1^H and ^13^C NMR resonances were fully assigned by the ^1^H–^1^H COSY, HSQC, ROESY, and HMBC experiments (Figure 3). Based on the above data, the structure of **1** was elucidated, as reflected in Figure 1, and named epiremisporine F.

Compound **2** was obtained as an amorphous powder. The ESI–MS showed the quasi-molecular ion [M + Na]^+^ at m/z 613, suggesting a molecular formula of C_31_H_26_O_12_, which was elucidated by the HR–ESI–MS (*m/z* 613.12928 [M + Na]^+^, calcd. 613.13219) and by the ^1^H and ^13^C NMR data. The IR spectrum showed the presence of hydroxyl (3460 cm^−1^), ester carbonyl (1748 cm^−1^), and conjugated carbonyl (1655 cm^−1^) groups. Compound **2** exhibited both ^1^H and ^13^C NMR signals as pairs in a ratio of 1:0.94 in CDCl_3_, indicating that **2** exists in a dynamic isomerism between major (2′*R*) and minor (2′*S*) isomers in CDCl_3_. The signal at δ 12.36 (2′*R*) and 12.41 (2′*S*) exhibited a chelated hydroxyl group with the carbonyl group. Comparison of the ^1^H and ^13^C NMR data of **2** with those of epiremisporine B [5], suggested that their structures were closely related, except that the 11-methoxy group of **2** replaced the 11-hydroxy group of epiremisporine B [5]. This was supported by both HMBC correlations between OMe-11 [δ_H_ 3.88 (2′*R*) and 3.91 (2′*S*)] and C-11 (δ_C_ 160.0) and ROESY correlations between OMe-11 [δ_H_ 3.88 (2′*R*) and 3.91 (2′*S*)] and H-10 [δ_H_ 6.57 (2′*R*) and 6.59 (2′*S*)]. The relative configuration of **2** was elucidated on the basis of ROESY experiments. The ROESY cross-peaks between H-3/H-4, H-3/H-3′, H-3/COOMe-2, COOMe-2′α/H-3′ (2′*S*), and COOMe-2′β/H_β_-4′ (2′*R*) suggested that H-3, H-4, H-3′, COOMe-2′ (2′*S*), and COOMe-2 were α-oriented, and COOMe-2′ (2′*R*) was β-oriented. To further confirm the relative configuration of **2**, a computer-assisted 3D structure was obtained by using the molecular-modeling program CS CHEM 3D Ultra 16.0, with MM2 force-field calculations for energy minimization. The calculated distances between H-3/H-4 (2′*S*) (2.117 Å), H-3/H-4 (2′*R*) (2.189 Å), H-3/H-3′ (2′*S*) (2.462 Å), H-3/H-3′ (2′*R*) (2.496 Å), COOMe-2′/H-3′ (2′*S*) (2.188 Å), COOMe-2′/H_β_-4′ (2′*R*) (3.744 Å), H-3/H-16 (2′*S*) (2.338 Å), and H-3/H-16 (2′*R*) (2.311 Å) were all less than 4 Å (Figure 4). This was consistent with the well-defined ROESY observed for each of these H-atom pairs (Figure 4). Compound **2** showed similar CD Cotton effects [207.5 (Δ*ε* –0.97), 220.0 (Δ*ε* +1.05), 234.0 (Δ*ε* –1.02), 257.5 (Δ*ε* +11.60), 281.5 (Δ*ε* –4.41), and 330.5 (Δ*ε* +4.27) nm] (Appendix A), compared with epiremisporine B [5]. Based on the above data, the structure of **2** was elucidated, as displayed in Figure 1 and Figure 5, and named epiremisporine G, which was further substantiated by the ^1^H-^1^H COSY, ROESY (Figure 5), HSQC, and HMBC (Figure 5) experiments.

Compound **3** was isolated as an amorphous powder. The ESI–MS displayed the quasi-molecular ion [M + Na]^+^ at m/z 627, suggesting a molecular formula of C_32_H_28_O_12_, which was elucidated by the HR–ESI–MS (*m/z* 627.14731 [M + Na]^+^, calcd. 627.14784) and by the ^1^H and ^13^C NMR data. The IR spectrum showed the presence of hydroxyl (3420 cm^−1^), ester carbonyl (1761 and 1740 cm^−1^), and the conjugated carbonyl (1657 cm^−1^) groups. The signals at δ 12.14 and 12.99 exhibited two chelated hydroxyl groups with the carbonyl group. Comparison of the ^1^H and ^13^C NMR data of **3** with those of epiremisporine F (**1**) suggested that their structures were closely related, except that the 4α-methyl group of **3** replaced the 4α-hydrogen of **1**. This was substantiated by both HMBC correlations between Me-4α (δ_H_ 2.12) and C-3 (δ_C_ 54.1), C-4 (δ_C_ 47.9), C-5 (δ_C_ 174.3), and C-14′ (δ_C_ 114.8), and ROESY correlations between Me-4α (δ_H_ 2.12) and H_α_-3 (δ_H_ 3.12) and H_α_-4′ (δ_H_ 3.08). The relative configuration of **3** was confirmed by ROESY experiments. The ROESY cross-peaks between Me-4α/H-3, Me-4α/H_α_-4′, H-3/H-3′, H-3/H_α_-4′, OMe-2′/H_β_-4′, and H-3/COOMe-2 suggested that H-3, Me-4, H-3′, and COOMe-2 were α-oriented, and OMe-2′ was β-oriented. To further confirm the relative configuration of **3**, a computer-assisted 3D structure was obtained by using the molecular-modeling program CS CHEM 3D Ultra 16.0, with MM2 force-field calculations for energy minimization. The calculated distances between H-3/Me-4 (2.161 Å), H-3/H-16 (2.320 Å), H-3/H-3′ (2.479 Å), and OMe-2′/H_β_-4′ (2.212 Å) were all less than 4 Å (Figure 6). This was consistent with the well-defined ROESY observed for each of these H-atom pairs (Figure 6). Compound **3** showed similar CD Cotton effects [207.5 (Δ*ε* +12.33), 229.5 (Δ*ε* −6.57), 263.0 (Δ*ε* +20.13), 290.5 (Δ*ε* −7.81), and 332.0 (Δ*ε* +6.57) nm], compared to the literature data [5]. Thus, **3** possessed a 2*S*,3*R*,2′*S*,3′*S*-configuration. The ^1^H and ^13^C NMR resonances were fully assigned by ^1^H–^1^H COSY, ROESY (Figure 7a), HSQC, and HMBC (Figure 7b) experiments. Based on the above data, the structure of **3** was elucidated, as displayed in Figure 1, and named epiremisporine H.

New compounds **1**–**3** were hypothesized to be biosynthesized from dimerization of their natural precursors, remisporine A [5], coniochaetone H [18], and (−)-preussochromone D [19] (Figure 8). The hypothetic biosynthesis schemes of **1**–**3** were proposed as shown in Figure 9 and Figure 10, respectively.

The correlations between the dihedral angles (H3′-C3′-C4′-H4′α and H3′-C3′-C4′-H4′β) and the vicinal coupling constants (*J*_3′, 4′α_ and *J*_3′, 4′β_) of compounds **1**–**3** and related analogues [5] are summarized in Table 1. The dihedral angles were calculated by using the molecular-modeling program CS CHEM 3D Ultra 16.0, with the MM2 force-field calculations for energy minimization. The correlations between dihedral angles (H3′-C3′-C4′-H4′α and H3′-C3′-C4′-H4′β) and vicinal coupling constants (*J*_3′, 4′α_ and *J*_3′, 4′β_) of compounds **1**–**3** were consistent with the Karplus relationship. The 2′*S*,3′*S*-configuration slightly decreased the *J*_3′, 4′β_ value from 11.3~12.7 to 8.4~11.7 compared to the 2′*R*,3′*S* configuration. These data could also support the structural confirmation of the new compounds **1**–**3**.

### 2.3. Biological Studies

#### 2.3.1. Inhibitory Activities on Neutrophil Pro-Inflammatory Responses

The anti-inflammatory activities of the isolates from *Penicillium citrinum* were evaluated by their ability to inhibit formyl-L-methionyl-L-leucyl-L-phenylalanine (fMLP)-induced O_2_^•–^ generation by human neutrophils. The anti-inflammatory activity data are shown in Table 2. The clinically used anti-inflammatory agent, ibuprofen [20,21,22,23], was used as the positive control. From the results of our anti-inflammatory tests, epiremisporine G (**2**) and epiremisporine H (**3**) exhibited inhibition (IC_50_ values ≤ 33.52 μM) of superoxide anion release by human neutrophils, in response to fMLP. Among the chromone derivatives, epiremisporine H (**3**) (with 4α-methyl and 2′β-methoxy groups) exhibited more effective anti-inflammatory activity than its analogues, epiremisporine C (with 4α-hydrogen and 2′α-methoxy group) [8] and epiremisporine F (with 4α-hydrogen and 2′β-methoxy group). In addition, epiremisporine B (with 11-hydroxyl group) [8] exhibited stronger anti-inflammatory activity than epiremisporine G (**2**) (with 11-methoxy group). Therefore, our study suggests *Penicillium citrinum* and its isolated compounds (**2**, **3**, and epiremisporine B) could be further discovered as promising candidates for the therapy or prevention of various inflammatory diseases.

#### 2.3.2. Cytotoxic Effects and Selectivity of Compounds **1**–**3**

In this study, the cytotoxic activities of three compounds (**1**–**3**) against HT-29 (human colon carcinoma) and A549 (human lung carcinoma) cells were studied, as shown in Table 3, Appendix A. 5-Fluorouracil (5-FU) was used as the positive control [24,25,26]. Among the isolated compounds, compounds **1**, **2**, and **3** exhibited potent cytotoxic activities with IC_50_ values of 44.77 ± 2.70, 35.05 ± 3.76, and 21.17 ± 4.89 μM against HT-29 cells, respectively. In addition, compounds **1**, **2**, and **3** exhibited cytotoxic activities with IC_50_ values of 77.05 ± 2.57, 52.30 ± 2.88, and 31.43 ± 3.01 μM against A549 cells, respectively. Among the chromone derivatives, epiremisporine H (**3**) (with 4-methyl and 11-hydroxyl groups) exhibited a more effective cytotoxic activity than its analogues, epiremisporines B–E [8], F, and G (without 4-methyl group) against the HT-29 and A549 cells. In other words, the new compound, epiremisporine H (**3**) (without 4α-H and with 4-Me & 11-OH groups), exhibited a stronger anticancer activity than its analogues, epiremisporines F and G (**1** and **2**) (with 4α-H and without 4-Me group) against HT-29 and A549 cells.

#### 2.3.3. New Compound **3** Inhibited Proliferation of HT-29 Cells

Epiremisporine H (**3**) exhibited the most potent cytotoxicity, with an IC_50_ value of 21.17 ± 4.89 μM against the HT-29 cell line. Compound **3** was selectively tested for clonogenic assay as it is a new compound and possesses cytotoxic activity against HT-29. The effect of compound **3** on colony formation of HT-29 cells was tested by using the clonogenic assay (Figure 11). The HT-29 cell colonies were visualized as blue discs, through crystal violet staining. It was clearly observed that compound **3** (12.5 μM) significantly reduced the colony formation of HT-29 cells. Moreover, compound **3** almost completely inhibited the colony formation at 25 μM. 

#### 2.3.4. Effects of Epiremisporine F (**3**) on Protein Expressions of Pro-caspase 3 and Cleaved Caspase 3 in HT-29 and A549 Cells

Caspase 3 activation is a hallmark of apoptosis. Caspase 3 activation involves the cleavage of pro-caspase 3 (the inactive precursor form of caspase 3), leading to the formation of cleaved caspase 3 (which is the active caspase 3). Upon apoptosis, the pro-caspase 3 would decrease and the cleaved caspase 3 would increase accordingly [27,28,29]. We further investigated whether epiremisporine F (**3**) was able to influence these enzymatic activities of caspase 3. The results show that compound **3** suppressed pro-caspase 3 and increased the cleaved caspase 3 (Figure 12 and Figure 13). Furthermore, compound **3** markedly induced apoptosis of HT-29 and A549 cells through caspase-3-dependent pathways.

#### 2.3.5. Effects of Compound **3** on Protein Expressions of Bax and Bcl-2 in HT-29 and A549 Cells

To determine whether compound **3** could influence the expression of proteins related to HT-29 and A549 cells apoptosis, compound **3** (6.25, 12.5, 25, and 50 μM) was added to HT-29 and A549 cells. Figure 12 and Figure 13 show that the expression level of pro-apoptotic protein bax was obviously higher with 50 μM treatment of compound **3** than with 6.25 or 12.5 μM treatment. On the contrary, the cells treated with 6.25 or 12.5 μM of compound **3** showed higher Bcl-2 (anti-apoptotic protein) expression than that treated with 50 μM. The results show that compound **3** suppressed the expression of Bcl-2 and increased bax expression.

## 3. Materials and Methods

### 3.1. General Procedures

Optical rotations were measured using a Jasco *P*-2000 polarimeter (Japan Spectroscopic Corporation, Tokyo, Japan) in CHCl_3_. Circular dichroism (CD) spectra were obtained on a J-715 spectropolarimeter (Jasco, Easton, MD, USA). Ultraviolet (UV) spectra were recorded on a Hitachi U-2800 Double Beam Spectrophotometer (Hitachi High-Technologies Corporation, Tokyo, Japan). Infrared (IR) spectra (neat or KBr) were obtained on a Shimadzu IRAffinity-1S FT-IR Spectrophotometer (Shimadzu Corporation, Kyoto, Japan). Nuclear magnetic resonance (NMR) spectra, including correlation spectroscopy (COSY), rotating frame nuclear Overhauser effect spectroscopy (ROESY), heteronuclear multiple-bond correlation (HMBC), and heteronuclear single-quantum coherence (HSQC) experiments, were acquired using a BRUKER AVIII-500 spectrometer (Bruker, Bremen, Germany), operating at 500 (^1^H) and 125 MHz (^13^C), respectively, with chemical shifts given in ppm (δ), using CDCl_3_ as an internal standard (peak at 7.263 ppm in ^1^H NMR and 77.0 ppm in ^13^C NMR spectrum). Electrospray ionization (ESI) and high-resolution electrospray ionization (HRESI) mass spectra were recorded on a Bruker APEX II Mass Spectrometer (Bruker, Bremen, Germany). Silica gel (70–230 mesh (63–200 μm) and 230–400 mesh (40–63 μm), Merck) was used for column chromatography (CC). Silica gel 60 F-254 (Merck, Darmstadt, Germany) was used for thin-layer chromatography (TLC) and preparative thin-layer chromatography (PTLC).

### 3.2. Fungal Material

The fungal strain *Penicillium citrinum* BCRC 09F458 was isolated from wastewater, which was collected from Hazailiao, Dongshi, Chiayi, Taiwan, in 2009. The fungal strain was identified as *Penicillium citrinum* (family Trichocomaceae) by the BCRC center, based on cultural and anamorphic data. The rDNA-ITS (internal transcribed spacer) region was used for further identification. After searching the GenBank database through BLAST (nucleotide sequence comparison program), it was found to have 100% similarity to *P. citrinum*. *P. citrinum* BCRC 09F458 was stored in the Biological Resources Collection and Research Center (BCRC) of the Food Industry Research and Development Institute (FIRDI).

#### Cultivation and Preparation of the Fungal Strain

*P. citrinum* BCRC 09F0458 was maintained on potato dextrose agar (PDA) and the strain was cultured on PDA at 25 °C for 7 days. The spores were seeded into 300 ml shake flasks containing 50 ml RGY (3% rice starch, 7% glycerol, 1.1% polypeptone, 3% soybean powder, 0.1% MgSO_4_, and 0.2% NaNO_3_), and cultivated with shaking (150 rpm) at 25°C for 3 days. After the mycelium enrichment step, an inoculum mixing 100 mL mycelium broth and 100 mL RGY medium was inoculated into plastic boxes (25 cm × 30 cm) containing 1.5 kg sterile rice and cultivated at 25 °C for producing rice, and the above-mentioned RGY medium was added for maintaining the growth. After 21 days of cultivation, the rice was harvested, and used as a sample for further extraction.

### 3.3. Extraction and Isolation

The rice of the *P. citrinum* BCRC 09F0458 (1.5 kg) was extracted with 95% EtOH (3 × 10 L, 3 d each) at room temperature. The ethanol extract was concentrated under reduced pressure, and was partitioned with *n*-BuOH/H_2_O (1:1, *v*/*v*) to afford *n*-BuOH soluble fraction (36.2 g), H_2_O soluble fraction (13.0 g), and insoluble fraction (500 mg). The *n*-BuOH fraction (fraction A, 36.2 g) was purified by column chromatography (CC) (1.6 kg of silica gel, 70–230 mesh (63–200 μm); *n*-hexane/EtOAc 25:1–0:1, 1500 mL) to afford 13 fractions: A1–A13. Fraction A9 (1.44 g) was subjected to MPLC (65 g of silica gel, 230–400 mesh (40–63 μm); dichloromethane/EtOAc 1:0–2:3, 650 mL fractions) to give 12 subfractions: A9-1–A9-12. Fraction A9-10 (89 mg) was further purified by semipreparative normal-phase HPLC (silica gel; *n*-hexane/EtOAc, 2:1) to afford epiremisporine H (**3**) (3.2 mg). Fraction A10 (0.98 g) was subjected to MPLC (44 g of silica gel, 230–400 mesh (40–63 μm); *n*-hexane/acetone 1:0–0:1, 450 mL fractions) to give 10 subfractions: A10-1–A10-10. Fraction A10-2 (96 mg) was further purified by preparative TLC (silica gel; *n*-hexane/dichloromethane/acetone, 3:1:1) to afford isoconiochaetone F (**1**) (4.1 mg). Fraction A11 (2.38 g) was subjected to MPLC (107 g of silica gel, 230–400 mesh (40–63 μm); *n*-hexane/acetone 1:0–0:1, 1000 mL-fractions) to give 14 subfractions: A11-1–A11-14. Fraction A11-8 (128 mg) was further purified by semipreparative normal-phase HPLC (silica gel; *n*-hexane/dichloromethane/EtOAc, 5:3:2) to afford epiremisporine G (**2**) (5.2 mg).

Epiremisporine F (**1**): [α${]}_{D}^{25}$ = +560.4° (*c* 0.13, CHCl_3_); UV (MeOH) λ_max_ nm (log ε): 241 (4.47), 327 (3.80) nm; ^1^H NMR data, see Table 4; ^13^C NMR data, see Table 5; HRESIMS, CD, 1D, and 2D NMR spectra, see Appendix A Appendix A.

Epiremisporine G (**2**): [α${]}_{D}^{25}$ = +522.8° (*c* 0.15, CHCl_3_); UV (MeOH) λ_max_ nm (log ε): 237 (4.43), 317 (3.83) nm; ^1^H NMR data, see Table 4; ^13^C NMR data, see Table 5; HRESIMS, CD, 1D, and 2D NMR spectra, see Appendix A Appendix A.

Epiremisporine H (**3**): [α${]}_{D}^{25}$ = +568.7° (*c* 0.11, CHCl_3_); UV (MeOH) λ_max_ nm (log ε): 240 (4.44), 327 (3.75) nm; ^1^H NMR data, see Table 4; ^13^C NMR data, see Table 5; HRESIMS, CD, 1D, and 2D NMR spectra, see Appendix A Appendix A.

### 3.4. Biological Assay

The anti-inflammatory effects of the isolated compounds from *Penicillium citrinum* were evaluated by suppressing fMLP-induced O_2_^•–^ generation by human neutrophils. In addition, anti-cancer activity was evaluated by cytotoxicity assay and Western blot analysis.

#### 3.4.1. Preparation of Human Neutrophils

These studies were performed according with the code of ethics of the world medical association for (declaration of Helsinki) experiments involving humans, and all protocols were in compliance with the Institutional Review Board (IRB) of National Yang Ming University (protocol code YM106033E-2 and date of approval: 10 April 2019). Human neutrophils from the venous blood [11] of healthy, adult volunteers (20–35 years old) were isolated using a standard method of dextran sedimentation, prior to centrifugation in a Ficoll Hypaque gradient and hypotonic lysis of erythrocytes, as previously described [30]. Purified neutrophils containing >98% viable cells, as determined by the trypan blue exclusion method, were resuspended in HBSS buffer at pH 7.4 and were maintained at 4 °C, prior to use [31].

#### 3.4.2. Measurement of O_2_^•–^ Generation

The assay for detection of O_2_^•–^ generation was based on the SOD-inhibitable reduction of ferricytochrome c [32]. In short, neutrophils (1 × 10^6^ cells/mL) pretreated with the various test agents (50 and 5 μM) at 37 °C for 5 min were stimulated with fMLP (1 μmol/L) in the presence of ferricytochrome c (0.5 mg/mL). Extracellular O_2_^•–^ production was evaluated with a UV spectrophotometer at 550 nm (Hitachi U-3010, Tokyo, Japan). The percentage of superoxide inhibition of the test compound was calculated as the percentage of inhibition = {(control − resting) − (compound − resting)}/(control − resting) × 100. The software SigmaPlot was used for determining the IC_50_ values [31].

#### 3.4.3. Chemicals and Antibodies

Fluorouracil (5-FU) and bovine serum albumin (BSA) were purchased from Sigma-Aldrich (St. Louis, MO, USA). The antibodies against Bcl-2, Bax, and β-actin were purchased from Cell Signaling Technology (Danvers, MA, USA). Caspase 3 was obtained from GeneTex International Corporation (Hsinchu, Taiwan).

#### 3.4.4. Cells and Culture Medium

HT-29 (human colon carcinoma) and A549 (human lung carcinoma) cells were kindly provided by Prof. Y. Su and Prof. T. M. Hu, respectively, of National Yang Ming Chiao Tung University, Taipei, Taiwan.

All cell lines were cultured in Dulbecco’s modified Eagle’s medium supplemented with 10% fetal bovine serum (FBS), 100 U/mL penicillin, 100 μg/mL streptomycin, 2 μM L-glutamine, and 1 mM sodium pyruvate. The cells were incubated in an atmosphere of 37 °C and 5% CO_2_ and passaged twice a week. Cells were stored in liquid nitrogen at −155 °C. After the cells were thawed, the experiment was completed before 30 generations. The purpose was to minimize experimental errors. The compound stock solution was stored in DMSO at a concentration of 10 mM and stored at −20 °C, and finally melted immediately before use.

#### 3.4.5. Cytotoxicity Assay

The cell viability was conducted by the MTT assay method, as previously described [33]. Briefly, 5 × 10^3^ cells in 200 μl per well were plated in 96-well culture plates and cultured in complete medium overnight. After 24 h, cells were treated with different concentrations (3.125, 6.25, 12.5, 25, 50, and 100 μM) of compounds **1**–**3**. Fluorouracil (5-FU) (Sigma-Aldrich, St. Louis, MO, USA) was used as a positive control against HT-29 and A549 cells. The optical density at 570 nm was measured by ELISA plate reader (μ Quant) and the IC_50_ value was calculated. The optical density of formazan formed in control (untreated) cells was taken as 100% viability.

#### 3.4.6. Clonogenic Assay

The clonogenic assay was determined by the reference method with a slight modification [34]. In this assay, HT-29 cells were seeded in 6-well plates with 3 × 10^3^ cells per well and incubated for 24 hours. The cells were then treated with the indicated concentrations of compound **3**, and cultured for 14 days. The cells were washed three times using PBS and fixed using 99% methanol for 30 min. After washing three times with distilled water, the cells were stained using 0.2% crystal violet dye for 15 min and rinsed with distilled water to wash away the excess dye. The visible colonies were compared with the control samples and photographed using a standard camera under natural light.

#### 3.4.7. Western Blotting Analysis

Western blot analysis was performed according to the method previously reported [8]. In brief, HT-29 (1.5 × 10^5^ cells) and A549 (1 × 10^5^ cells) were seeded into 6-well plates and grown until 85–90% confluent. Then, different concentrations (6.25, 12.5, 25, and 50 μM) of compound **3** was added. Cells were collected and lysed by radioimmunoprecipitation assay (RIPA) buffer. Lysates of total protein were separated by 12.5% sodium dodecyl sulfate-polyacrylamide gels and transferred to polyvinylidene difluoride (PVDF) membranes. After blocking, the membranes were incubated with anti-Bax, anti-Bcl-2 (Cell Signaling Inc., Danvers, MA, USA), anti-caspase-3, and anti-β-actin (GeneTex Inc., Irvine, CA, USA) primary antibodies at 4 °C overnight. Then, each membrane was incubated with horseradish peroxidase (HRP)-conjugated secondary antibodies at room temperature, for 1 h, while shaking. At last, each membrane was excited using an enhanced chemiluminescence (ECL) detection kit, and the images were visualized by ImageQuant LAS 4000 Mini biomolecular imager (GE Healthcare, Woburn, MA, USA). The band densities were quantified using the ImageJ software (NIH, Bethesda, MD, USA).

#### 3.4.8. Statistical Analysis

All results are presented as mean ± SEM. Statistical analysis was executed by using Student’s *t*-test. A probability of 0.05 or less was considered to be statistically significant. Microsoft Excel 2019 was used for the statistical and graphical assessment. All experiments were executed at least 3 times.

## 4. Conclusions

Three novel compounds (**1**–**3**) were isolated and identified from *Penicillium citrinum*. The structures of these compounds were established on the basis of spectroscopic data. Reactive oxygen species (ROS) (e.g., superoxide anion (O_2_^•−^), hydrogen peroxide) produced by human neutrophils contribute to the pathogenesis of inflammatory diseases. Among the isolated compounds, compounds **2** and **3** could inhibit fMLP-induced O_2_^•−^ generation, with IC_50_ values ≤ 33.52 μM. These isolated compounds are worth further research, as promising new leads for developing anti-inflammatory agents. Furthermore, compound **3** markedly induced apoptosis of HT-29 cells through the mitochondrial- and caspase-3-dependent pathways (Figure 14). This suggests that compound **3** is worth further investigation and might be developed as a candidate for the treatment or prevention of colon cancer.

## Figures and Tables

**Figure 1 marinedrugs-19-00408-f001:**
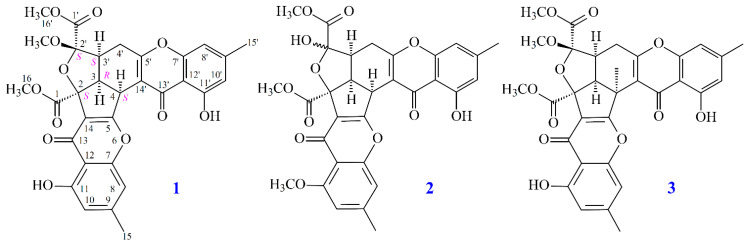
The chemical structures of compounds **1**–**3** isolated from *Penicillium citrinum*.

**Figure 2 marinedrugs-19-00408-f002:**
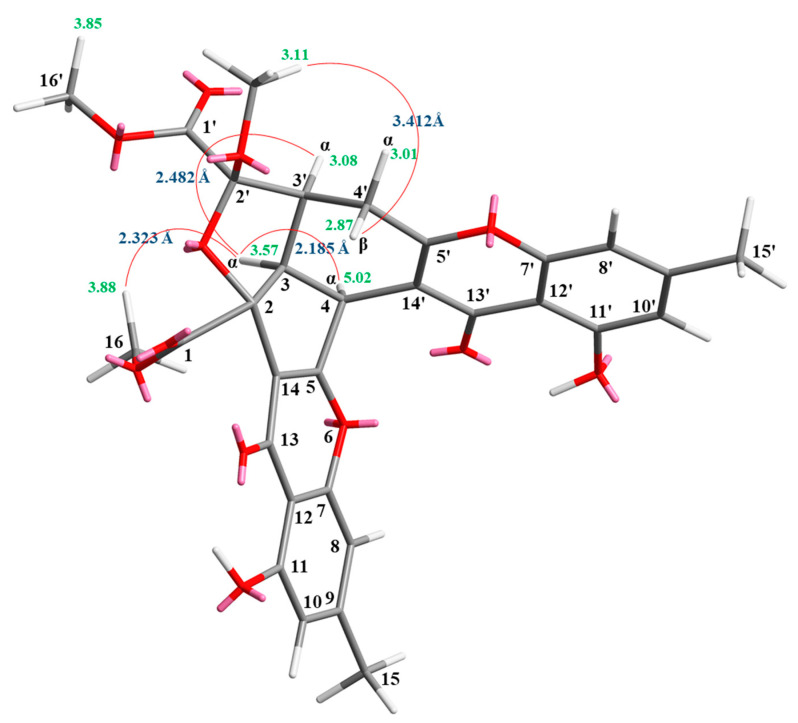
Selected ROESY correlations and relative configuration of **1**.

**Figure 3 marinedrugs-19-00408-f003:**
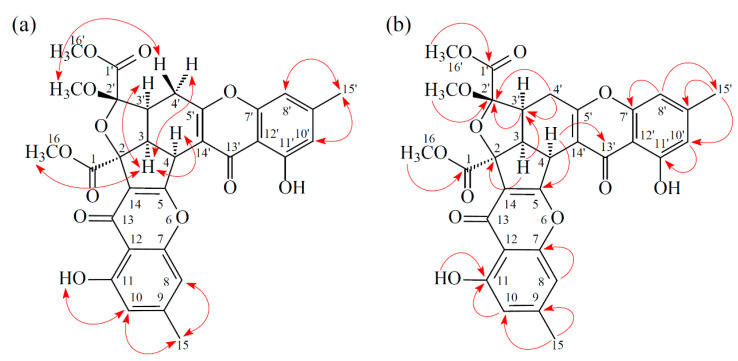
Key ROESY (**a**) and HMBC (**b**) correlations of **1**.

**Figure 4 marinedrugs-19-00408-f004:**
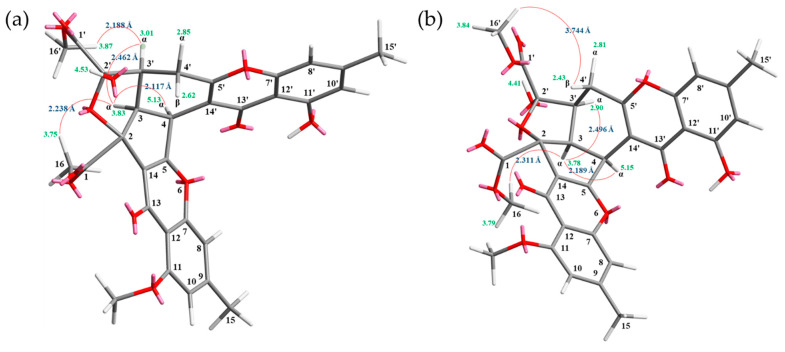
Selected ROESY correlations and relative configuration of **2** (2’*S*) (**a**) and **2** (2’*R*) (**b**).

**Figure 5 marinedrugs-19-00408-f005:**
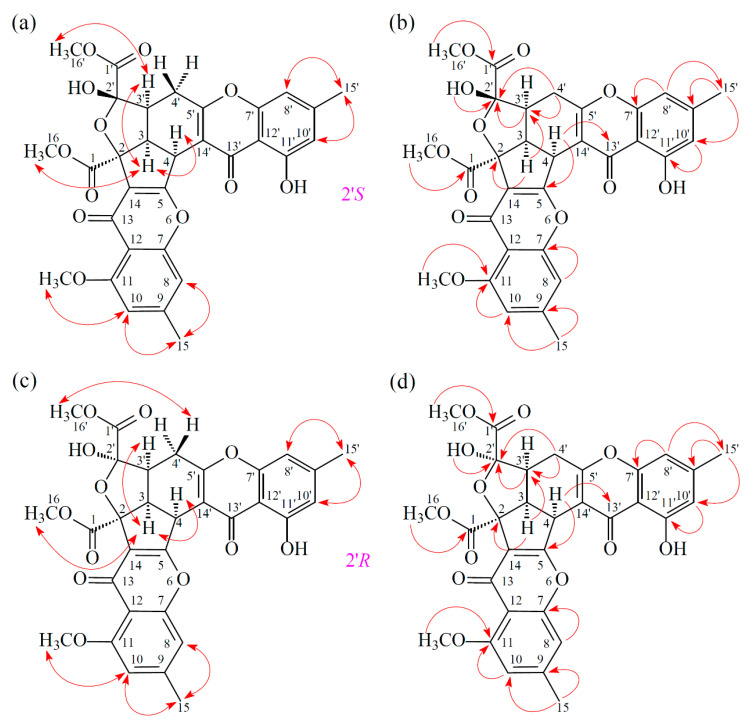
Key ROESY (**a**) and HMBC (**b**) correlations of **2** (2′*S*). Key ROESY (**c**) and HMBC (**d**) correlations of **2** (2′*R*).

**Figure 6 marinedrugs-19-00408-f006:**
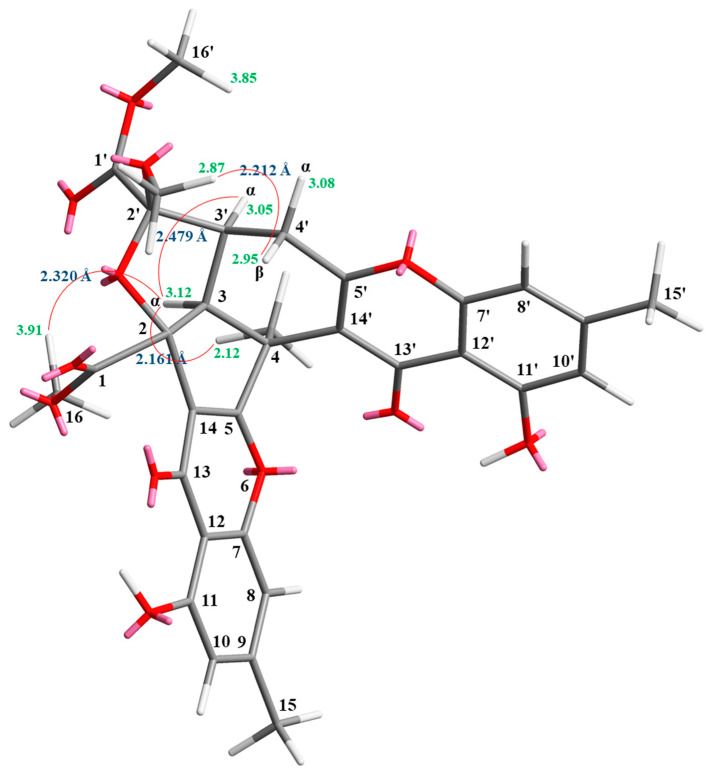
Selected ROESY correlations and the relative configuration of **3**.

**Figure 7 marinedrugs-19-00408-f007:**
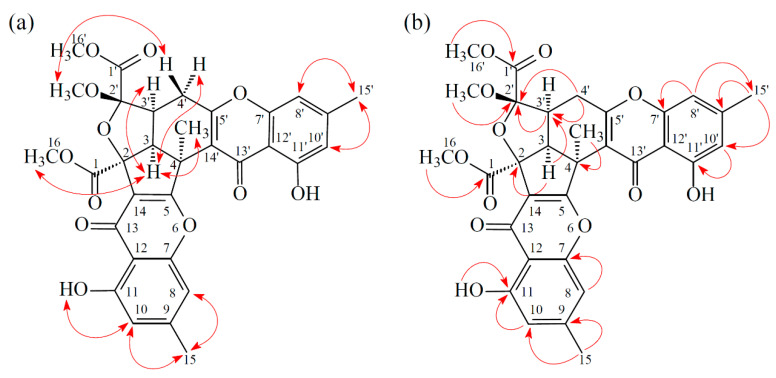
Key ROESY (**a**) and HMBC (**b**) correlations of **3**.

**Figure 8 marinedrugs-19-00408-f008:**
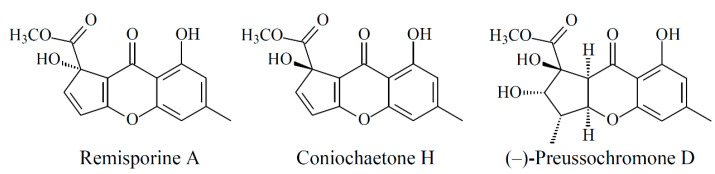
The chemical structures of remisporine A, coniochaetone H, and (−)-preussochromone D.

**Figure 9 marinedrugs-19-00408-f009:**
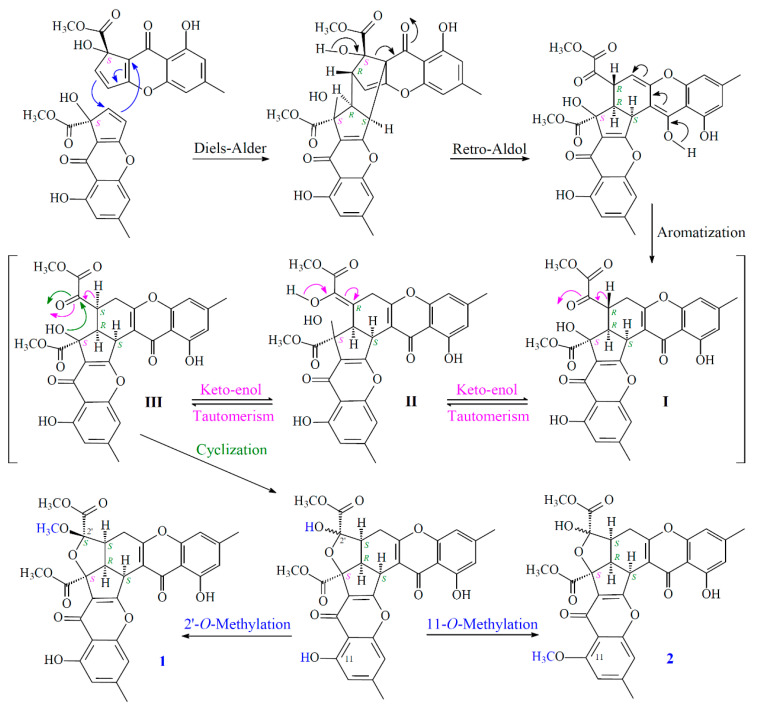
The hypothetic biosynthesis scheme of **1** and **2**.

**Figure 10 marinedrugs-19-00408-f010:**
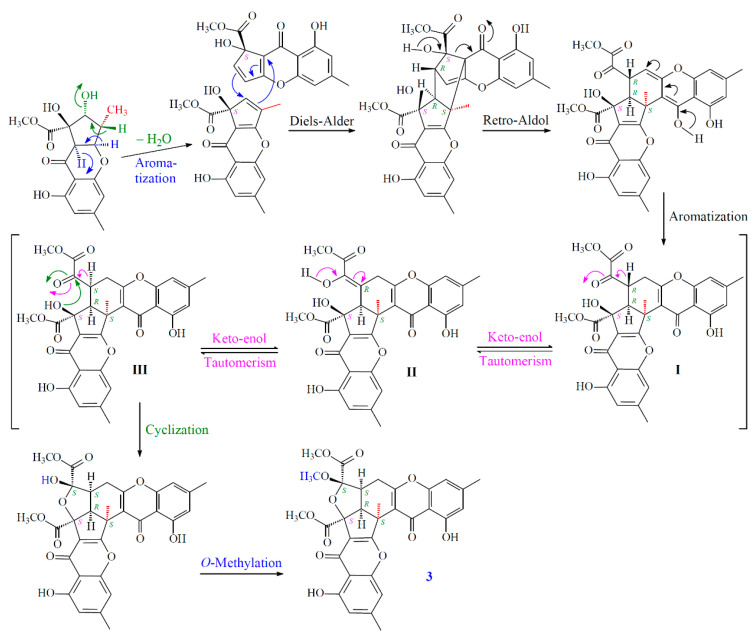
The hypothetic biosynthesis scheme of **3**.

**Figure 11 marinedrugs-19-00408-f011:**
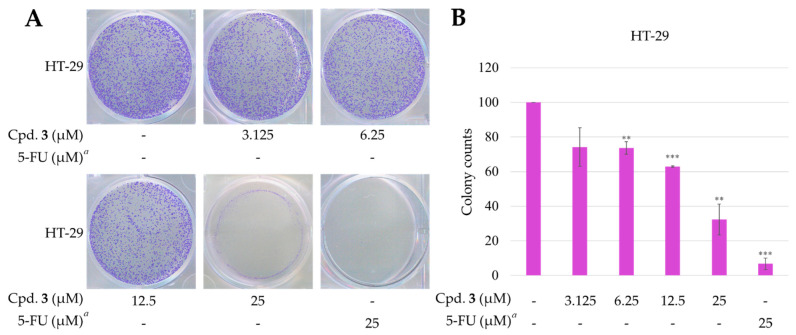
Effect of epiremisporine H (**3**) on the colony formation of HT-29 cells. (**A**) The effect of **3** against HT-29 cell colony formation. Clonogenicity was assessed by the monolayer colony formation assay. Representative images show the blue colonies of HT-29 cells stained with crystal violet. (**B**) Histogram presentation of HT-29 cell colony quantification. ** *p* < 0.01; *** *p* < 0.001 compared with the control. *^a^* 5-FU (5-fluorouracil) was used as a positive control. Cpd. **3** means compound **3** (epiremisporine H).

**Figure 12 marinedrugs-19-00408-f012:**
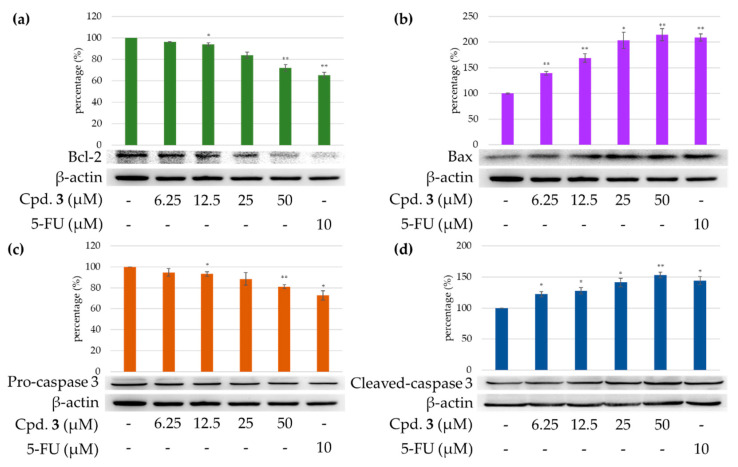
Immunoblot analysis for Bcl-2 (**a**), Bax (**b**), pro-caspase 3 (**c**), and cleaved caspase 3 (**d**) in each group on HT-29 cells. Treatment with epiremisporine F (**3**) significantly decreased the expression levels of Bcl-2 and pro-caspase 3, and raised the expression levels of Bax and cleaved caspase 3. Asterisks indicate significant differences (* *p* < 0.05 and ** *p* < 0.01) compared with the control group. Cpd. **3** means compound **3** (epiremisporine H).

**Figure 13 marinedrugs-19-00408-f013:**
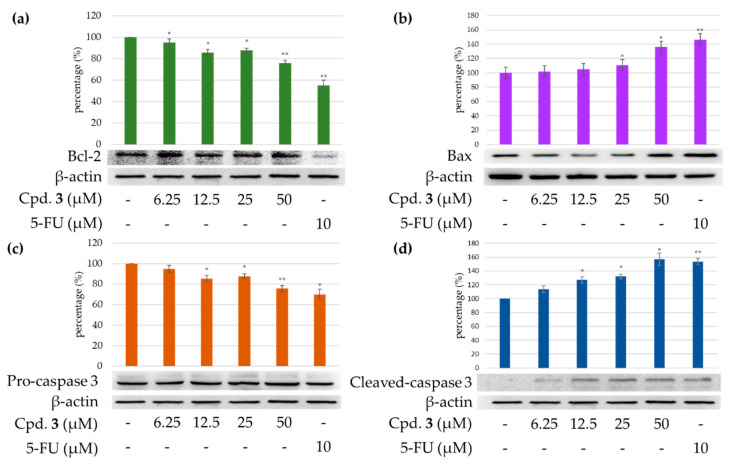
Immunoblot analysis for Bcl-2 (**a**), Bax (**b**), pro-caspase 3 (**c**), and cleaved caspase 3 (**d**) in each group on A549 cells. Treatment with epiremisporine F (**3**) significantly decreased the expression levels of Bcl-2 and pro-caspase 3, and raised the expression levels of Bax and cleaved caspase 3. Asterisks indicate significant differences (* *p* < 0.05 and ** *p* < 0.01) compared with the control group. Cpd. **3** means compound **3** (epiremisporine H).

**Figure 14 marinedrugs-19-00408-f014:**
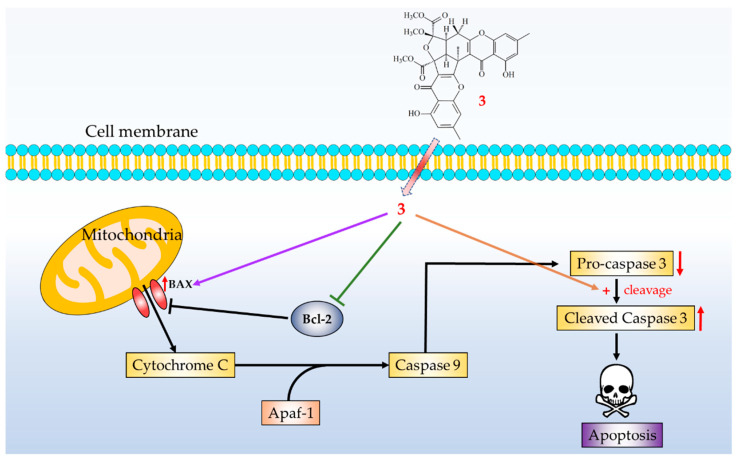
Schematic diagram for cancer cell apoptosis mechanism of compound **3** in HT-29 cells.

**Table 1 marinedrugs-19-00408-t001:** The correlations between dihedral angles and vicinal coupling constants of compounds **1**–**3** and related analogues [5].

Compounds	Dihedral Angles	*J*_3′, 4′α_ (Hz)	Dihedral Angles	*J*_3′, 4′β_ (Hz)
	(H3′-C3′-C4′-H4′α)		(H3′-C3′-C4′-H4′β)	
**1** (2′*S*,3′*S*)	54.5°	4.7	173.7°	8.4
**2** (2′*R*,3′*S*)	54.3°	4.7	173.9°	12.7
**2** (2′*S*,3′*S*)	54.3°	5.9	174.0°	11.5
**3** (2′*S*,3′*S*)	55.2°	5.9	175.7°	8.4
Epiremisporine B (2′*R*,3′*S*)	53.9°	5.4	173.5°	12.7
Epiremisporine B (2′*S*,3′*S*)	54.7°	5.9	173.8°	11.7
Epiremisporine B1 (2′*R*,3′*S*)	54.2°	6.6	173.8°	11.3
Epiremisporine B1 (2′*S*,3′*S*)	56.0°	6.5	175.2°	10.3
Remisporine B (2′*S*,3′*R*)	178.8°	12.2	61.0°	4.3

**Table 2 marinedrugs-19-00408-t002:** Inhibitory effects of compounds **1**–**3** from *Penicillium citrinum* on superoxide anion generation by human neutrophils, in response to fMLP.

Compounds	IC_50_ (μM) ^a^
Epiremisporine F (**1**)	>50
Epiremisporine G (**2**)	31.68 ± 2.53 ^c^
Epiremisporine H (**3**)	33.52 ± 0.42 ^c^
Ibuprofen ^b^	28.56 ± 2.73 ^c^

^a^ Concentration necessary for 50% inhibition (IC_50_). ^b^ Ibuprofen (a fMLP receptor antagonist) was used as a positive control. Results are presented as average ± SEM (*n* = 3). Values are expressed as average ± SEM (*n* = 3). ^c^ *p* < 0.01 compared with the control.

**Table 3 marinedrugs-19-00408-t003:** Cytotoxic effects of compounds **1**–**3** against A549 and HT-29 cells.

Compounds	IC_50_ (μM) ^a^	
	HT-29	A549
Epiremisporine F (**1**)	44.77 ± 2.70 ^c^	77.05 ± 2.57 ^c^
Epiremisporine G (**2**)	35.05 ± 3.76 ^d^	52.30 ± 2.88 ^d^
Epiremisporine H (**3**)	21.17 ± 4.89 ^e^	31.43 ± 3.01 ^d^
5-FU ^b^	17.47 ± 1.67 ^e^	10.57 ± 1.89 ^d^

^a^ The IC_50_ values were calculated from the slope of dose–response curves (SigmaPlot). Values are expressed as mean ± SEM (*n* = 3). ^c^ *p* < 0.05; ^d^ *p* < 0.01; ^e^ *p* < 0.001 compared with the control. ^b^ 5-Fluorouracil (5-FU) was used as a positive control.

**Table 4 marinedrugs-19-00408-t004:** ^1^H NMR data (500 MHz, CDCl_3_) for **1**–**3**.

Position	1	2 (2’*S*)	2 (2’*R*)	3
	δ_H_ (*J* in Hz)			
3	3.57 (dd, 10.5, 9.1)	3.83 (t, 9.0)	3.78 (dd, 9.0, 8.7)	3.12 (d, 10.3)
4	5.02 (d, 9.1)	5.13 (d, 9.0)	5.15 (d, 9.0)	-
8	6.71 (br s)	6.78 (br s)	6.77 (br s)	6.80 (br s)
10	6.61 (br s)	6.59 (br s)	6.57 (br s)	6.62 (br s)
15	2.35 (s)	2.37 (s)	2.36 (s)	2.38 (s)
16	3.88 (s)	3.75 (s)	3.79 (s)	3.91 (s)
3′	3.08 (ddd, 10.5, 8.4, 4.7)	3.01 (ddd, 11.5, 9.0, 5.9)	2.90 (ddd, 12.7, 8.7, 4.7)	3.05 (ddd, 10.3, 8.4, 5.9)
4′α	3.01 (dd, 18.7, 4.7)	2.85 (dd, 16.7, 5.9)	2.81 (dd, 15.8, 4.7)	3.08 (dd, 18.9, 5.9)
4′β	2.87 (dd, 18.7, 8.4)	2.62 (dd, 16.7, 11.5)	2.43 (dd, 15.8,12.7)	2.95 (dd, 18.9, 8.4)
8′	6.70 (br s)	6.71 (br s)	6.70 (br s)	6.64 (br s)
10′	6.68 (br s)	6.69 (br s)	6.69 (br s)	6.63 (br s)
15′	2.42 (s)	2.42 (s)	2.41 (s)	2.40 (s)
16′	3.85 (s)	3.87 (s)	3.84 (s)	3.85 (s)
11-OH	12.19 (s)	-	-	12.14 (s)
11-OMe	-	3.91 (s)	3.88 (s)	-
2-OH	-	4.53 (br s)	4.41 (s)	-
4-Me	-	-	-	2.12 (s)
2′-OMe	3.11 (s)	-		2.87 (s)
11′-OH	12.42 (s)	12.41 (s)	12.36 (s)	12.99 (s)

**Table 5 marinedrugs-19-00408-t005:** ^13^C NMR data (125 MHz, CDCl_3_) for **1**–**3**.

Position	1	2 (2’*S*)	2 (2’*R*)	3
	δ_C_, Type			
1	170.2, C	171.3, C	172.8, C	170.6, C
2	91.1, C	89.2, C	91.3, C	90.5, C
3	44.0, CH	48.2, CH	47.0, CH	54.1, CH
4	37.8, CH	36.7, CH	35.9, CH	47.4, CH
5	169.9, C	165.3, C	165.1, C	174.3, C
7	157.2, C	159.2, C	159.2, C	157.4, C
8	108.5, CH	110.7, CH	110.7, CH	108.9, CH
9	147.2, C	145.4, C	145.2, C	147.0, C
10	113.0, CH	108.3, CH	108.4, CH	112.7, CH
11	160.8, C	160.0, C	160.0, C	160.6, C
12	108.7, C	112.7, C	112.8, C	108.7, C
13	179.2, C	173.8, C	173.7, C	179.6, C
14	118.5, C	121.9, C	121.2, C	117.0, C
15	22.2, CH_3_	22.1, CH_3_	22.1, CH_3_	22.2, CH_3_
16	53.3, CH_3_	53.0, CH_3_	53.5, CH_3_	53.4, CH_3_
1′	168.2, C	170.0, C	167.5, C	167.6, C
2′	107.9, C	104.6, C	105.9, C	107.6, C
3′	43.3, CH	43.2, CH	48.3, CH	42.3, CH
4′	25.4, CH_2_	26.3, CH_2_	27.5, CH_2_	25.5, CH_2_
5′	165.4, C	166.7, C	166.1, C	165.1, C
7′	155.9, C	156.1, C	156.1, C	155.1, C
8′	107.4, CH	107.5, CH	107.6, CH	106.8, CH
9′	147.4, C	147.4, C	147.5, C	147.3, C
10′	112.3, CH	112.6, CH	112.6, CH	112.0, CH
11′	160.3, C	160.5, C	160.5, C	160.8, C
12′	108.3, C	108.5, C	108.5, C	108.7, C
13′	180.5, C	179.9, C	179.8, C	181.3, C
14′	111.3, C	112.7, C	112.4, C	114.8, C
15′	22.4, CH_3_	22.4, CH_3_	22.4, CH_3_	22.3, CH_3_
16′	52.9, CH_3_	53.2, CH_3_	52.9, CH_3_	52.9, CH_3_
11-OMe	-	56.3, CH_3_	56.3, CH_3_	-
4-Me	-	-	-	28.4, CH_3_
2′-OMe	52.3, CH_3_	-	-	51.3, CH_3_

## Data Availability

The data presented in this study are available in the main text and the supplementary materials of this article.

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
