# Peer review of "Anti-Cancer and Anti-Inflammatory Activities of Three New Chromone Derivatives from the Marine-Derived Penicillium citrinum"

_marinedrugs, 2021, doi:10.3390/md19080408_

Round 1
Reviewer 1 Report
This manuscript describes the isolation and bioactivity of new chromone derivatives from the marine-derived Penicillium citrinum.
It seems to me that the description of structural elucidation and bioactivity of new natural products epiremisporine F (1), epiremisporine G (2), and epiremisporine H (3) is interesting,
And the manuscript including table, figures and reference is well organized.
However, Chromone Derivatives epiremisporine C, epiremisporine D, and epiremisporine E were reported in this journal ‘Marine Drugs’, titled 'Rare Chromone Derivatives from the Marine-Derived Penicillium citrinum with Anti-Cancer and Anti-Inflammatory Activities’ by same authors. (Marine Drugs 2021 Jan 8;19(1):25)
For this reason, it seems to me that this manuscript can't be accepted by the readers of this journal. I would recommend sending it in another journal related to natural products.
Reviewer 2 Report
The article concerns the isolation and structural elucidation of three new and uncommon chromone derivatives, namely epiremisporine F, epiremisporine G and epiremisporine H from marine-originated Penicillium citrinum. The compounds remarkably suppressed fMLP-induced superoxide anion generation by human neutrophils. Epiremisporine H exhibited cytotoxic activities on human colon carcinoma (HT-29) and non-small lung cancer cell (A549). Moreover, it was found that this substance induced apoptosis of tumor cells via Bcl-2, Bax, and caspase 3 signaling cascades. The structures of all the substances that were very similar to each other were elucidated using 2D NMR, HR-mass-spectrometry and even the computation the stereochemistry by molecular modelling. The biotesting results are adequate presented and the substances may be effective candidates for antiinflammation preparations.
1) The article is clearly written but the introductive and conclusive parts should be improved. There is too many general information but not any introductive information concerning the discussed class of natural products and its distribution amongst fungi. The introduction should be expanded but the general information may be shortened.
2) The English in Introduction seems to be awkward and should be also corrected.
3) The differences in the number of carbon atoms in 1 and 2 on the one hand and 3 on the other hand (the presence of a methyl at C-4 in 3 instead of H at C-4 in 1 and 2) should be explained from biosynthetic point of view. The hypothetic biosynthesis scheme explained formation 1, 2 and 3 should be created and inserted.
My general opinion: the article is interesting and may be published after minor revision.
Reviewer 3 Report
The article entitled "Anti-Cancer and Anti-Inflammatory Activities of Three New Chromone Derivatives from the Marine-Derived Penicillium citrinum" is worth to read and well-planned study. However, there are some issues to address before consider to publish in marine drugs.
First issue is the references in this article. The authors not careful to add sufficient references for results and discussion section. Specifically, authors are failed to compare their results with previous studies and justify their results. So, I recommend to revise results and discussion section.
Please provide evidence for Ibuprofen for its clinical applications
Why you use 5-FU for this study. Please justify (in the discussion part)
Please provide reference for this statement (p9)
Caspase 3 activation is a hallmark of apoptosis. Caspase 3 activation involves the cleavage of pro-caspase 3 (the inactive precursor form of caspase 3), leading to the for-mation of cleaved-caspase 3 (which is the active caspase 3). Upon apoptosis, the pro-caspase 3 would decrease and the cleaved-caspase 3 would increase accordingly
In addition, authors add large number (18) of references to just for 2 sentences in the introduction part and some are not related to those statements. Please recheck your references.
Please provide dose range determination data for the anti-inflammatory study.
Recheck the fig 10.b quantification data. At the 12.5um concentration bax expression is very low. But quantification data showing upregulation compared to the 6.25. Please replace both bands with more describable bands
Please provide cytotoxicity determination data of cancer cells as supplementary data (representative data for table 3).
The IC 50 for 3rd compound is 21.17 ± 4.89 against HT29 cells (Table 3). But according to the figure 8 @25 um it has less than 25% colony counts according to the authors (according to my observation almost nothing compared to control). Please explain these results. I recommend to repeat this results
Define Cpd. 3
Round 2
Reviewer 3 Report
I have no further comments regarding this manuscript as the authors address all questions at a satisfactory level. Therefore, I recommend this manuscript to publish in Marine Drugs